# ParallelEdits: Efficient Multi-Aspect Text-Driven Image Editing with Attention Grouping

**Mingzhen Huang, Jialing Cai, Shan Jia, Vishnu Suresh Lokhande∗, Siwei Lyu∗**
University at Buffalo, State University of New York, USA

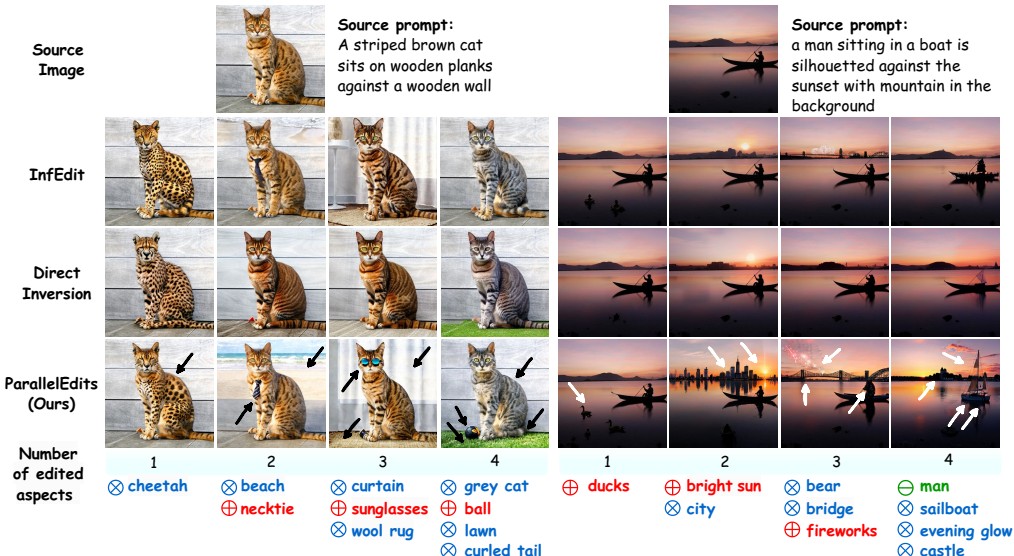

Figure 1: **Multi-aspect text-driven image editing.** Multiple edits in images pose a significant challenge in existing models (such as DirectInverison [1] and InfEdit [2]), as their performance downgrades with an increasing number of aspects. In contrast, our ParallelEdits can achieve precise multi-aspect image editing in 5 seconds. The symbol ⊗ denotes a swap action, the symbol ⊕ denotes an object addition action, and the symbol ⊖ denotes an object deletion. Arrows (→) on the image highlight the aspects edited by our method.

## Abstract

Text-driven image synthesis has made significant advancements with the development of diffusion models, transforming how visual content is generated from text prompts. Despite these advances, text-driven image editing, a key area in computer graphics, faces unique challenges. A major challenge is making simultaneous edits across multiple objects or attributes. Applying these methods sequentially for multi-aspect edits increases computational demands and efficiency losses. In this paper, we address these challenges with significant contributions. Our main contribution is the development of ParallelEdits, a method that seamlessly manages simultaneous edits across multiple attributes. In contrast to previous approaches, ParallelEdits not only preserves the quality of single attribute edits but also significantly improves the performance of multitasking edits. This is achieved through innovative attention distribution mechanism and multi-branch design that operates across several processing heads. Additionally, we introduce the PIE-Bench++ dataset, an expansion of the original PIE-Bench dataset, to better support evaluating image-editing tasks involving multiple objects and attributes simultaneously. This dataset is a benchmark for evaluating text-driven image editing methods in multifaceted scenarios. Codes are available at: https://mingzhen-huang.github.io/projects/ParallelEdits.html.

∗Corresponding authors

38th Conference on Neural Information Processing Systems (NeurIPS 2024).

# 1 Introduction

Recently, text-driven image editing has experienced remarkable growth, driven by advances in diffusion-based image generative models. This technique involves modifying existing images based on textual prompts to alter objects, their attributes, and the relationships among various objects. The latest methods [3, 1, 4] can produce edited images that closely match the semantic content described in the prompts while keeping the rest of the image unchanged. Unlike early image editing approaches that required image matting to precisely extract foreground objects using alpha mattes [5], text-driven editing offers a less labor-intensive alternative. User-provided textual prompts guide the edits, with auxiliary inputs like masks facilitating localized modifications [6].

While these methods have showcased promising results, existing methods typically focus on editing a single aspect in the source image. An "aspect" refers to a specific attribute or entity within the textual prompt that describes the image and can be modified, such as object type, color, material, pose, or relationship. However, the ability to edit multiple aspects through text prompts is rarely explored. We introduce the concept of *multi-aspect text-driven image editing* to address this gap. Multi-aspect image editing is essential due to the rich and diverse content and structure of digital images, as well as the varied requirements of users. For example, it always occurs that users wish to modify multiple attributes or regions in an image, such as adding a necktie to a cat and changing the background wall to a beach (Fig. 1, Left), or removing a man and replacing a mountain with a castle in the right example. Unlike traditional editing methods (e.g., [1, 2]) that focus on a single aspect, multi-aspect editing allows users to manipulate various aspects simultaneously. Different from full text-to-image synthesis [7, 8], which involves creating content from scratch, multi-aspect editing works with the source image to ensure essential content preservation. It bridges the gap between single-aspect editing and full synthesis, catering to a wide range of editing scenarios.

However, we observe that directly applying the single-aspect text-driven image editing methods in cases where multiple image aspects must be modified often does not yield satisfactory results. A straightforward solution to this problem is to apply the single aspect editing method *sequentially* – we can order the aspects to be modified and use a single-aspect editing method to change the aspects one by one. Although sequential applications of single-aspect text-driven image editing methods can modify multiple aspects of an image, they may introduce significantly higher computational overhead. More importantly, the order of the aspects modified may affect the quality – changes to later aspects may undo the early ones or accumulate the errors and artifacts, thus reducing the effectiveness of the final editing results, as the last two rows of Fig. 5 and Table 1 show.

In this work, we introduce *ParallelEdits* as an efficient and effective solution to the problem of multi-aspect text-driven image editing. This method is based on a crucial insight that the editing step can occur in parallel with the image's diffusion steps. Therefore, in ParallelEdits, we build image aspect editing into the diffusion steps to accelerate the editing process. ParrallelEdits is based on an architecture with a fixed number of additional branches dedicated to handling rigid, non-rigid, and style changes. This design ensures scalability independent of the number of prompt aspects altered. In addition, we employ an attention aggregator to accurately assess editing difficulty and route aspects to appropriate branches within the ParallelEdits framework, ensuring precise and efficient editing. To enable subsequent research and evaluation of multi-aspect text-driven image editing methods, we also build the PIE-Bench++ dataset, an extension of the PIE-Bench [1] that has 700 images with detailed text prompts and tailored to facilitate simultaneous edits across multiple image aspects. We propose evaluation metrics and benchmark different text-driven image editing methods on PIE-Bench++. The ParallelEdits outperforms the state-of-the-art image editing methods on PIE-Bench++.

# 2 Related Works

**Diffusion Models for Text-Driven Image Editing**. Text-driven image editing aims to manipulate local regions of an image based on textual prompts. The editing has two main goals: ensuring the edits align with provided instructions and preserving essential content. Diffusion models [9] have gained popularity as a preferred image editing model for their capacity for generating high-quality samples by incorporating diverse conditions, especially using text [10, 11, 2, 12–14, 1]. This involves transforming the images into the latent space and generating regions using diffusion models conditioned by the text prompt while ensuring accurate reconstruction of unmodified regions

during editing. To avoid the edited image deviating from original image, early text-driven image editing typically requires user-specified masks as additional condition [15–17] or training [18–20] to guided the editing process, which constrain their potential zero-shot application. To address this limitation, recent editing models, such as InfEdit [2], PnP [21], Direct Inversion [1] follow the work Prompt-to-Prompt (P2P) [3], which proposed to obtain an attention map from the cross attention process and either swap or refine the attention map from text prompt for image editing. This design automatically obtains the editing mask and only allows image editing using a text prompt. Another method, MasaCtrl [4], converts existing self-attention in diffusion models into mutual self-attention for non-rigid consistent image synthesis and editing, enabling to query correlated local contents and textures from source images for consistency.

**Multi-Aspect Image Editing**. While current image editing models have shown promising results in their text-driven image editing benchmarks, we have observed that they work well on single-attribute editing while struggling to edit multiple aspects, especially when editing multiple objects (as shown in Fig. 1). We attribute this limitation to the following reasons. First, existing methods use the attention mask to direct where edits should be made. With multiple attributes, the editing area may expand significantly, incorporating extensive semantic information or scattered regions that are challenging to edit using a single mask. Second, employing a fixed mask from cross-attention maps struggles with edits involving changes in region size (such as pose adjustments), while using an adaptive mask faces challenges in maintaining edit fidelity. Therefore, integrating various attention masks for accurate multi-attribute editing presents a challenging technical problem. Early studies [22, 23] have employed GAN models such as StyleGAN2 [24] to edit multiple attributes in faces. The multiple-attribute editing is realized by training the GAN model with supervised multi-class training and a training dataset of image and attribute vector pairs. This solution heavily relies on the training sets and has limitations in generalizing to new editing types. Few recent works achieve multi-aspect editing with additional inputs: [25] leverages rich text to edit multiple objects and [26] pre-processes the image with grounding to localize multiple edited regions for multi-aspect editing. However, the editing performance highly relies on additional input beyond plain text, either from user input or other off-the-shelf models. A recent work [27] proposes an iterative multi-granular image editor, where a diffusion model can faithfully follow a series of image editing instructions from a user. However, this interactive editing pipeline will result in significant computational overhead.

**Image Editing with Multiple Branches.** In the literature [4, 3], image editing processes have been conducted by implementing a dual-branch approach. This methodology involves segregating source and target branches throughout the editing process. Specifically, the source branch is reverted to $z_0$, while the trajectory of the target branch is iteratively adjusted. By computing the distance from the source branch, the calibration of the target branch occurs at each time-step. Our observation underscores the disparity between the effectiveness of a dual branch in enhancing the editing process and its failure in multi-aspect editing. A singular target branch proves inadequate in calibrating fully from the source branch, leading to imperfect incorporation of all aspects into the image. Hence, our primary proposition advocates for multi-aspect editing by utilizing multiple target branches. Each target branch's trajectory is meticulously calibrated, with simpler concepts addressed in the initial branches and more complex aspects deferred to subsequent ones. In the following section, we will delve deeper into this concept.

## 3 Diffusion-based Image Generation and Editing

We are provided with an image sample $x_0$ which transforms the latent space via an encoder/decoder pair $\mathcal{E}/\mathcal{D}$, such that $z_0 = \mathcal{E}(x_0)$. Here, $z_0$ represents the latent representation of the image $x_0$. With a slight abuse of notation, we approximate the reconstructed image $\bar{x}_0$ as $\mathcal{D}(\bar{z}_0)$, where $\bar{z}_0$ denotes the reconstructed version of $z_0$. These operations are integral to the latent diffusion model [9]. The diffusion process constitutes two steps: the forward step incrementally adds zero-mean white Gaussian noise with time-varying variance to the latent vector $z$ according to discrete-time $t^*$,

$$z_t = \sqrt{\alpha_t} z_0 + \sqrt{1 - \alpha_t} \epsilon \quad \text{with} \quad \epsilon \sim \mathcal{N}(0, I), \tag{1}$$

$\alpha_{1:T}$ represents a variance schedule for $t$ drawn from the interval $[1, T]$. The variance schedule can be different, such as linear or cosine quadratic [28]. The backward step is an iterative process to remove

---

*Diffusion process is rigorously defined as a continuous-time stochastic differential equation, but in practice often implemented with discrete-time updates.

the noise from the data progressively. Using the same variance schedule $\alpha_{1:T}$ as in the forward step, a noise schedule $\sigma_{1:T}$ and a parameterized noise prediction network $\epsilon_\theta$ with coefficients $c_{\text{pred}} = \sqrt{\alpha_{t-1}}$, $c_{\text{dir}} = \sqrt{1 - \alpha_{t-1} - \sigma_t^2}$, and $c_{\text{noise}} = \sigma_t$, the backward step corresponds to the following process:

$$z_{t-1} = \underbrace{c_{\text{pred}} f_\theta(z_t, t)}_{\text{predicting } \bar{z}_0} + \underbrace{c_{\text{dir}} \epsilon_\theta(z_t, t)}_{\text{adjust along } z_t} + \underbrace{c_{\text{noise}} \epsilon_t}_{\text{random noise}} \quad \text{with} \quad \epsilon_t \sim \mathcal{N}(0, I) \tag{2}$$

The noise schedule $\sigma_{1:T}$ comprises hyperparameters requiring careful selection based on factors like image dimensions or desired performance [29][30]. In the framework of Denoising Diffusion Implicit Models (DDIM) [31], the function $f_\theta$ is employed for the prediction and reconstruction of $\bar{z}_0$, based on the input $z_t$. Specifically, we have $\bar{z}_0 = f_\theta(z_t, t) = \frac{1}{\sqrt{\alpha_t}} z_t - \frac{\sqrt{1-\alpha_t}}{\sqrt{\alpha_t}} \epsilon_\theta(z_t, t)$.

**Consistency Models for Inversion-free Image Editing**. Consistency models [32, 33] have been introduced to expedite the generation process through a consistent distillation approach. These models exhibit a self-consistency property, ensuring that samples along the same trajectory map to the same initial point. Specifically, the function $f_\theta$ is rendered self-consistent by satisfying $f_\theta(z_t, t) = z_0$ for a given sample $z_t$ at timestep $t$. As a result, the self-consistency property yields a closed-form solution for the noise predictor $\epsilon_\theta$. We denote this particular $\epsilon_\theta$ as $\epsilon^{\text{cons}}$, which is derived as $\epsilon^{\text{cons}} = \frac{z_t - \sqrt{\alpha_t} z_0}{\sqrt{1-\alpha_t}}$. Since $\epsilon^{\text{cons}}$ is not parameterized and contains the ground-truth $z_0$, Xu *et al.* [2] propose starting directly with random noise, i.e., $z_T \sim \mathcal{N}(0, \mathbf{I})$, at the last time-step $T$, which is particularly advantageous for image-editing tasks as it eliminates the need for inversion from $z_0$ to $z_T$. Therefore, starting with $z_\tau = z_T \sim \mathcal{N}(0, \mathbf{I})$, the sampling process proceeds as follows:

① $z = \frac{z_\tau - \sqrt{1-\alpha_\tau} \epsilon_\tau^{\text{cons}}}{\sqrt{\alpha_\tau}}$. Where, $\epsilon_\tau^{\text{cons}}$ is given by $\frac{z_\tau - \sqrt{\alpha_t} z_0}{\sqrt{1-\alpha_t}}$

② Noise is added to $z_\tau$, i.e, $z_\tau = \sqrt{\alpha_\tau} z + \sqrt{1 - \alpha_\tau} \epsilon$ where $\epsilon \sim \mathcal{N}(0, \mathbf{I})$

After many iterations, the final output is $z$. Furthermore, [2] demonstrates that the dual-branch paradigm (involving a source and a target branch) used in image editing tasks can be executed in an inversion-free manner. We will delve into this, along with our method description, in Section 4.2.2.

# 4 Multi-Aspect Image Editing

## 4.1 Problem Definition

The input to the multi-aspect image editing task includes a source image ($\mathcal{I}_{src}$), the source prompt, and a set of edits to be applied to the source image, indicating the changes from the source prompt to target prompt. A text prompt (whether source or target) comprises several independent tokens, of which only a subset is editable. We refer to these editable tokens as *Aspects*.

**Definition 4.1** (Aspect). We define an $i^{\text{th}}$ aspect $\mathcal{A}_{src}^i$ in the source prompt (or the $j^{\text{th}}$ aspect $\mathcal{A}_{edt}^j$ in the target prompt) as any entity that can be substituted, deleted, or inserted into the text prompt, resulting in a meaningful sentence structure.

Several examples of tokens corresponding to aspects or not are given in Fig. 3. In other words, aspects correspond to single or multiple tokens representing object color, pose, material, content, background, image style, etc. An editing operation $E^{i \to j}$ between the editing pair $(\mathcal{A}_{src}^i, \mathcal{A}_{edt}^j)$ as $E^{i \to j} \in \{\otimes, \oplus, \ominus, \oslash\}$. Here, $\otimes$ denotes a swap action, $\oplus$ denotes an object addition action, $\ominus$ denotes object deletion, and $\oslash$ indicates no change in the aspect. Such an editing operation can be inferred directly by appropriately mapping the source and target prompts, or it can be provided as metadata [3, 34]. The editing task is considered successful if the edited source image, $\mathcal{I}_{edt}$, reflects the required edits while preserving the unaffected aspects of the original image.

## 4.2 Method

Figure 2 outlines the overall pipeline of our method, which has three steps. In the first step (Sec. 4.2.1), we perform *aspect grouping* using attention maps generated by running a few iterations of the diffusion process. The aspects in the source image are put into up to $N$ groups, each processed by a distinct branch. The second step (Sec. 4.2.2) demonstrates how each branch, which receives a specific group of aspects, performs inversion-free editing. In the last step (Sec. 4.2.3), we perform the necessary adjustments for enabling cross-branch interaction and elucidate the significance of such interaction.

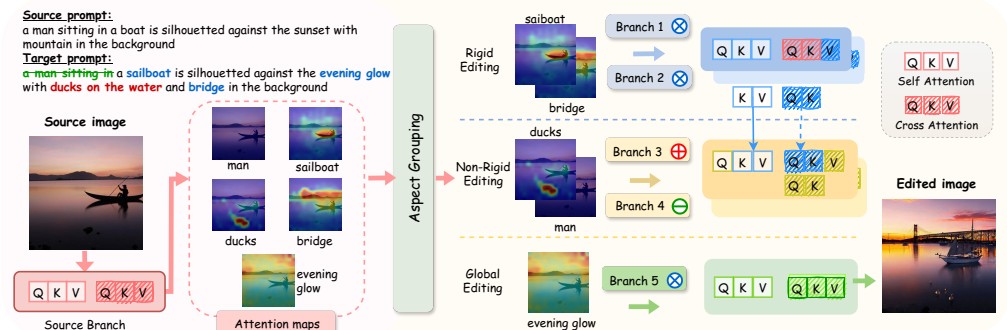

Figure 2: **Pipeline.** Our method, ParallelEdits, takes a source image, source prompt, and target prompt as input and produces an edited image. The target prompt specifies the edits needed in the source image. Attention maps for all edited aspects are first collected. Aspect Grouping (see Section 4.2.1) categorizes each aspect into one of $N$ groups (in the above figure, $N = 5$). Each group is then assigned a branch and the branch-level updates are detailed in Section 4.2.2. Each branch can be viewed either as a rigid editing branch, non-rigid editing branch, or global editing branch. Finally, adjustments to query/key/value at the self-attention and cross-attention layers are made, as illustrated in the figure and described in Section 4.2.3.

### 4.2.1 Aspect Grouping

We would like to group aspects in a prompt into $N$ distinct groups using the cross-attention maps of the diffusion UNet [35] to characterize the spatial layouts as in previous studies [36]. Given an editing operation $E^{i \to j}$ between the source aspect $\mathcal{A}^i_{src}$ and the target aspect $\mathcal{A}^j_{edt}$, we obtain the corresponding attention maps from both the source and target prompts as $\bar{\mathcal{M}}^i_{src}$ and $\bar{\mathcal{M}}^j_{edt}$, respectively. The attention map $\mathcal{M}$ is defined by the query feature $\hat{Q}$ and key feature $\hat{K}$ from the cross-attention as $\mathcal{M} = \text{softmax}\left(\frac{\hat{Q}\hat{K}^T}{\sqrt{d}}\right)$. The binarized attention map $\bar{\mathcal{M}}$ is obtained by normalizing $\mathcal{M}$ and thresholding its values. Our aspect grouping proceeds in two steps,

**Step 1. Assign a type for every editing operation ($E^{i \to j}$).** We consider three possible types of edits, in line with previous works [4], namely a global edit, a local rigid edit or a local non-rigid edit. Rigid local edits, such as changing an object's color or texture, do not alter the layout of objects. Conversely, non-rigid local edits modify the layout of objects, such as adding or deleting objects or changing object poses. Global edits affect background and style changes. The type assignment for the editing operation ($E^{i \to j}$) is determined by the following rules:

Figure 3: **Aspects and Aspect Grouping.** In a text prompt, there are multiple independent tokens, with only some being editable, known as aspects and are underlined in the above example. These aspects can be added, deleted, or swapped between the source and target prompts. Pairs of source and target aspects are grouped into branches, and the methodology for aspect grouping is explained in Section 4.2.1.

$$\text{type}(E^{i \to j}) = \begin{cases} \text{global edit} & \dots\dots\dots\dots\dots\dots\dots\dots\dots\dots\dots\dots\dots\dots\dots\dots \gamma(\bar{\mathcal{M}}^j_{edt}) \geq \beta\gamma\left(\sum\{\bar{\mathcal{M}}_{edt}\}\right) \\ \text{non-rigid edit} & \phi(\bar{\mathcal{M}}^i_{src}, \bar{\mathcal{M}}^j_{edt}) < \lambda \\ \text{rigid edit} & \phi(\bar{\mathcal{M}}^i_{src}, \bar{\mathcal{M}}^j_{edt}) \geq \lambda \end{cases} \Bigg\} \text{local edit} \dots \gamma(\bar{\mathcal{M}}^j_{edt}) < \beta\gamma\left(\sum\{\bar{\mathcal{M}}_{edt}\}\right) \quad (3)$$

Here, $\phi$ represent mIoU [37], while $\gamma$ indicates the alpha mattes of attention maps. $\lambda$ and $\beta$ are tunable hyperparameters. For further details, please refer to the supplementary Sec. D.

**Step 2. Categorize every editing operation ($\mathbf{E^{i \to j}}$) into N groups.** For each editing operation ($E^{i \to j}$) of a specific type, we assess whether $\phi(\bar{\mathcal{M}}^j_{edt}, \bar{\mathcal{M}}^k_{edt}) \geq \lambda$ to determine if there exists substantial overlap between any pair of attention maps of that type. If significant overlap is detected, the attention maps are grouped together. On the other hand, if attention maps are isolated like the "boat" and "mountain" in Fig. 3 are categorized into separate groups due to small overall. Therefore, we have a total of $N$ groups. Each group has a dedicated branch, resulting in a total of $N > 2$ branches.

### 4.2.2 Inversion-Free Multi-Branch Editing

We use a set of $N$ branches indexed by $n$. These $N$ branches are in addition to a source branch (also shown in Figure 2) that undergoes a DDCM sampling process [2]. The $n^{\text{th}}$ branch is calibrated to its $(n-1)^{\text{th}}$ branch, and the first branch is calibrated to the source branch. The $N-$way target branch calibration can occur simultaneously, saving significant compute time. For the DDCM sampling process of the $n^{\text{th}}$ branch, it has the form of Section 3, Step ①:

$$\overbrace{z(n)^{\text{edt}} \quad = \Big( \quad z(n)^{\text{edt}}_\tau \quad - \sqrt{1 - \alpha_\tau}\big(\epsilon(n)^{\text{edt}}_\tau - \epsilon(n-1)^{\text{edt}}_\tau + \quad \epsilon(n)^{\text{cons}}_\tau \quad \big)\Big)/\sqrt{\alpha_\tau}}^{\text{Updating } n^{\text{th}} \text{ branch}} \quad (4)$$

$$\underset{\text{edited latent}}{} \quad \underset{\text{noisy latent}}{} \quad \underset{\text{parameterized noise}}{} \quad \underset{\text{consistency noise}}{}$$

Let us break down Eq. 4 step by step. $n = 1$ representing the source branch, we have $z(1)^{\text{edt}} = z^{\text{src}}$ and $\epsilon(1)^{\text{edt}}_\tau = \epsilon^{\text{src}}_\tau$. Also, $z(1)^{\text{edt}}_\tau = z^{\text{src}}_\tau$, which at time step $\tau = \tau_1$, is random noise drawn from $\mathcal{N}(0, \mathbf{I})$. Similarly, when $n = N$, $z(N)^{\text{edt}}$ represents the final calibrated/edited image containing all the required aspect edits after repeating for $\tau \in \{\tau_1, \tau_2, \ldots \tau_T\}$ timesteps. The noise addition on any target branch remains the same as Step ②, i.e., $z(n)^{\text{edt}}_\tau = \sqrt{\alpha_\tau} z(n)^{\text{edt}} + \sqrt{1 - \alpha_\tau}\epsilon$ where $\epsilon \sim \mathcal{N}(0, \mathbf{I})$. For $1 < n < N$, we have $\epsilon(n)^{\text{edt}}_\tau = \epsilon_\theta(z(n)^{\text{edt}}_\tau, \tau)$, where $\epsilon_\theta$ represents a parameterized noise predictor network (details in the Appendix Sec. D). A key observation is that the difference in the parameterized noise at the $n^{\text{th}}$ branch and $(n-1)^{\text{th}}$ branch is utilized to calculate $z(n)^{\text{edt}}$ in (4). Finally, $\epsilon(n)^{\text{cons}}_\tau$ is defined by $\epsilon(n)^{\text{cons}}_\tau = (z(n)^{\text{edt}}_\tau - \sqrt{\alpha_\tau}\hat{z}(n-1)^{\text{edt}})/\sqrt{1 - \alpha_\tau}$. Unlike the dual-branch setup in [2], the reference initial input is the estimated latent from the previous branch at a previous diffusion denoising iteration as indicated by $\hat{z}(n-1)^{\text{edt}}$.

### 4.2.3 Cross-Branch Interactions

For **rigid local branches**, the cross-attention map $\mathcal{M}^i_n$ from the previous branch is either switched or injected into the current branch, akin to the method used in P2P [3]. This approach facilitates local edits while preserving structural consistency. For **non-rigid local branches**, we observe that the query features in the shallow layers of UNet [35] can effectively query correlated local contents and textures from the prior branch's latent features, ensuring consistency. Consequently, the key and value features from the prior branch are retained in the current branch to maintain consistent editing. We use a non-rigid editing branch to manage non-rigid local edits. In the current branch $n$, textures from the previous branch $(n-1)$ are preserved by replacing the $K_{n-1}$ and $V_{n-1}$ features from the last branch with the $K_n$ and $V_n$ features in the current branch. Only the query features are preserved to maintain layout semantic correspondence. Additionally, the attention mask $\mathcal{M}_{n-1}$ from the previous branch's cross-attention layer is used to guide the editing process by adding it to $\mathcal{M}_n$, thereby converting the object layout from $\mathcal{M}_{n-1}$ to $\mathcal{M}_n$. This step is crucial for object removal or shape modification edits, where the object mask is derived from the previous branch. For all **global branches**, there is no replacement of attention features or masks, and the attention mask is not used to guide the editing process, as the entire image is intended to be altered.

## 5 Experiments

**PIE-Bench++ Dataset**. We introduce a new dataset, PIE-Bench++, derived from PIE-Bench [1] and dedicated to evaluate the performance of multi-aspect image editing. The PIE-Bench dataset contains 700 images and prompts with single-aspect editing including object-level manipulations (addition,

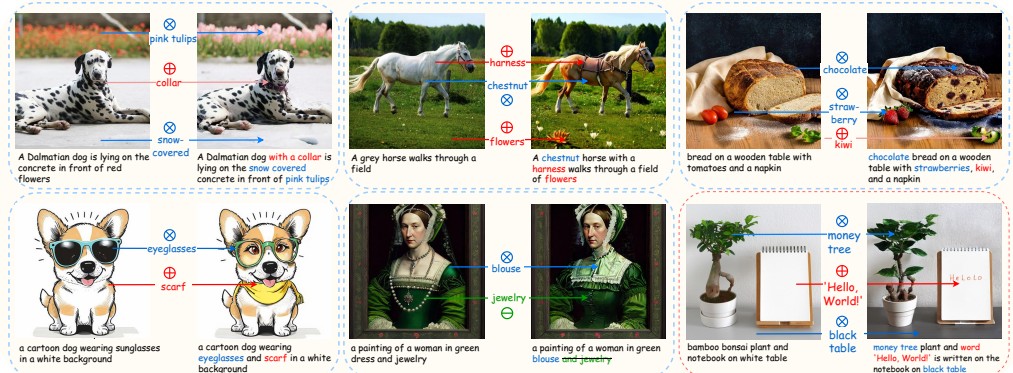

Figure 4: **Qualitative results of ParallelEdits.** We denote the edits in arrows with edit actions and aspects for each pair of images. The last image pair is a failure case of ParallelEdits.

deletion, or alteration), attribute-level manipulations (changes in content, pose, color, and material), and image-level manipulations that modify background and overall style. Our PIE-Bench++ extends PIE-Bench by enabling multi-aspect edits: 57% of our dataset have two aspect edits per prompt, 19% have more than two edits, and the remaining 24% have a signle aspect edit. For additional details and examples of the PIE-Bench++ dataset, please refer to the supplementary material.

**Evaluation Metrics**. We introduce two new metrics designed for evaluating multi-aspect text-driven image editing, alongside standard evaluation metrics.

**(a) Aspect Accuracy-LLaVA**. Drawing inspiration from the remarkable capability of large vision language models in comprehending intricate semantics within images, we propose to innovatively leverage them as an "omniscient" agent equipped with extensive knowledge to understand various attributes of images. We use the LLaVA [38] model, trained on visual grounding tasks, to evaluate the accuracy of multi-aspect image editing. Given a text prompt with multiple aspects, such as "*A [pink] [taxi] with [colorful] [flowers] on top*", we provide the following prompt with the edited image to the LLaVA model: "*Does the image match the elements in [ ]: A [pink] [taxi] with [colorful] [flowers] on top? Return a list of numbers where 1 is matched and 0 is unmatched.*" We then parse the returned list and compute its average to determine the aspect accuracy. We name this new evaluation metric as *AspAcc-LLaVA*. Examples and detailed explanations of this evaluation metric are available in the supplementary material.

**(b) Aspect Accuracy-CLIP**. We also use the similarity of the CLIP [39] to evaluate if an attribute has been successfully edited. Given an edited image $\mathcal{I}_{edt}$ and the target prompt $\mathcal{P}_{edt}$ with $k$ edited aspects $\mathcal{A}_{edt}$, every time we remove an aspect $\mathcal{A}_{edt}^{j}$ from $\mathcal{P}_{edt}$ and revert it back to $\mathcal{A}_{src}^{i}$ as $\hat{\mathcal{P}}_{edt}$. We then extract the CLIP [39] similarity between the edited image $I_{edt}$ and two prompts, i.e., $s_1 = CLIP(\mathcal{I}_{edt}, \mathcal{P}_{edt})$ and $s_2 = CLIP(\mathcal{I}_{edt}, \hat{\mathcal{P}}_{edt})$. We expect $s_1 > s_2$ if the aspect $\mathcal{A}_{edt}^{j}$ has been successfully edited. Thus, the aspect accuracy is $\frac{k_s}{k}$ when a total of $k_s$ aspects have been successfully edited among $k$ target edits. Note that in the case of an edited or added object that also involves changes in attributes (such as color or material), we consider it a successful edit only if both the object and its attributes have been successfully modified. We name this metric as *AspAcc-CLIP*.

**(c) Standard Metrics**. Several standard metrics widely used for evaluating text-image similarity and image quality are considered, including PSNR, LPIPS [40], MSE, and SSIM [41]. We also use the CLIP [39] score to measure the image-text alignment performance. Additionally, the bi-directional CLIP (D-CLIP) score [42] is reported, which is formulated as follows:

$$\cos\langle\text{CLIP}_{\text{img}}\left(\mathcal{I}_{edt}\right) - \text{CLIP}_{\text{img}}\left(\mathcal{I}_{src}\right), \text{CLIP}_{\text{text}}\left(\mathcal{P}_{edt}\right) - \text{CLIP}_{\text{text}}(\mathcal{P}_{src})\rangle$$

## 5.1 Quantitative Results

We first conduct experiment on the PIE-Bench++ dataset to compare our method with the state-of-the-art text-driven image editing methods combining their corresponding inversion method leads to best

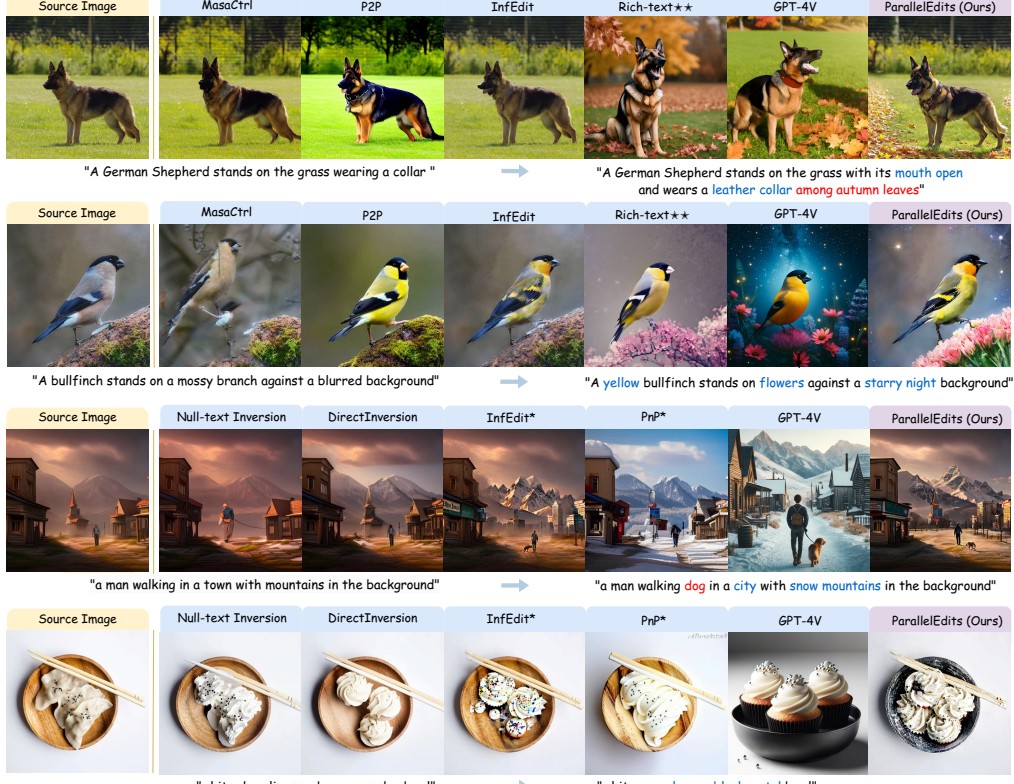

Figure 5: **Qualitative results comparison.** Current methods fail to edit multiple aspects effectively, even using sequential edits (noted as *). Methods marked with ★★ taking additional inputs other than source image and plain text.

| | StyleD | MasaCtrl | P2P | DI | NTI | InfEdit | PnP | DI* | P2P* | InfEdit* | PnP* | Ours |
|---|---|---|---|---|---|---|---|---|---|---|---|---|
| CLIP (%) ↑ | 24.02 | 23.37 | 24.00 | 24.40 | 24.03 | 24.44 | 24.90 | 22.80 | 25.13 | 25.17 | 25.39 | **25.70** |
| D-CLIP (%) ↑ | 8.43 | 7.68 | 11.43 | 13.23 | 12.08 | 11.02 | 11.83 | 2.74 | 8.30 | 11.77 | 11.85 | **20.70** |
| Eff. (secs/sample) ↓ | 382.98 | 12.70 | 33.72 | 29.70 | 145.29 | **2.22** | 32.51 | 100.98 | 121.32 | 11.82 | 122.81 | 4.98 |
| AspAcc-CLIP (%) ↑ | 32.37 | 34.05 | 26.14 | 31.95 | 42.19 | 42.38 | 44.91 | 28.23 | 38.96 | 42.38 | 48.20 | **51.05** |
| AspAcc-LLaVA (%) ↑ | 53.79 | 55.79 | 55.04 | 54.42 | 59.80 | 60.55 | 61.36 | 46.24 | 55.21 | 61.90 | 63.80 | **65.19** |

Table 1: **Comparison results in multi-aspect image editing on the PIE-Bench++ dataset.** Computational efficiency is abbreviated as Eff., and * denotes the method using sequential editing. The best performance is highlighted in **bold** and the second best performance is underlined.

performance, including DDIM+MasaCtrl [4], DDIM+Prompt-to-Prompt (P2P) [3], DDIM+Plug-and-Play (PnP) [21], StyleDiffusion (StyleD) [43]+P2P, Null-text Inversion (NTI) [34]+P2P, DirectInverison (DI)[1]+PnP, and InfEdit [2]. An intuitive way to improve off-the-shelf image editing methods is to apply the single-aspect editing method sequentially. We follow [27] to adapt existing image editing methods into sequential editing processes, where these methods are applied multiple times to achieve multi-aspect editing. Each time, only one aspect is edited. Table 1 presents the metrics in terms of text-image similarity (i.e., CILP and D-CLIP scores), computational efficiency, and aspect accuracy. Our ParallelEdits model outperforms all baselines in editing effectiveness, with a slightly longer runtime than the InfEdit model. Even though sequential editing better aligns the target prompt than their vanilla methods, it significantly increases computational overhead and may propagate editing errors over time. Moreover, although the sequential editing is conducted in the latent space, it would introduce more noise and artifacts to the edited image. Hence, their performance in all editing quality metrics was inferior to our method.

## 5.2 Qualitative Results

Fig. 4 presents several examples of our method's multi-aspect editing on the PIE-Bench++ dataset. The results demonstrate the effectiveness of our method in handling multiple and varied types of

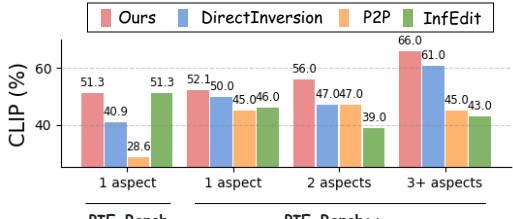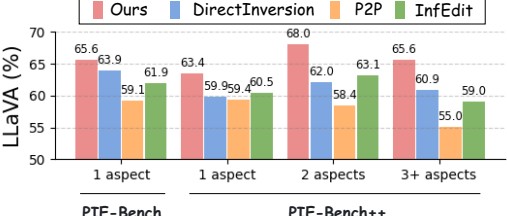

Figure 6: **Comparison across different numbers of editing aspects.** We also include the comparison in PIE-Bench dataset. Our proposed method is robust to different numbers of editing aspects.

| Methods | Background Preservation | | | | Aspect Preservation% | |
| | PSNR↑ | LPIPS$_{\times 10^3}$↓ | MSE$_{\times 10^4}$↓ | SSIM$_{\times 10^2}$↑ | CLIP↑ | LLaVA↑ |
|---|---|---|---|---|---|---|
| P2P [3] | 18.48 / 16.64 | 188.26 / 231.83 | 190.07 / 345.07 | 73.55 / 69.17 | 20.72 / 23.48 | 66.59 / 72.60 |
| PnP [21] | 22.73 / 21.54 | 103.16 / 120.87 | **75.97** / 102.47 | 80.73 / 78.85 | 20.79 / **25.59** | 75.65 / 78.77 |
| InfEdit [2] | 24.61 / 24.09 | 103.99 / 107.43 | 160.54 / 163.72 | 78.85 / 79.64 | 24.69 / 25.04 | 75.90 / 78.05 |
| Ours | **26.13** | **95.87** | 113.86 | **82.35** | 25.49 | **80.70** |

Table 2: **Comparison results in terms of background and aspects preservation.** The results from sequential editing is noted as green. ParallelEdits achieves state-of-the-art performance on multi-aspect editing while preserving the background and content consistency.

edits across diverse image content. Fig. 5 further compares our method with several state-of-the-art models and one popular multi-modal large language model, GPT-4V [44], by providing the source image, source prompt, and target prompt to guide the image editing. The Rich-text [25] model differs from other models, which uses rich-text prompt to edit the image generated from the plain (source) text prompt. The results show that current image editing models even with sequential editing fail to edit multiple aspects, while multi-modal large language models fail to preserve the content of source image. Our method achieves visually convincing results by successfully editing different attributes with good content preservation.

## 5.3 Ablation Study and Analysis

**(a) Impact of Editing Aspect Number.** We first examine the performance of our ParallelEdits and baseline methods on various editing aspect numbers by comparing CLIP and LLaVA-based aspect accuracies on the original PIE-Bench [1] and our PIE-Bench++ datasets. The bar charts in Fig. 6 show the outstanding performance of our method across all settings, including single-aspect editing on two datasets and multi-aspect editing. *Takeaway: the proposed ParallelEdits demonstrates robustness across varying numbers of editing aspects.*

**(b) Evaluation on Perservation.** We follow [1] to evaluate the background preservation. We first use the PSNR, LPIPS [40], MSE and SSIM [41] to evaluate the background preservation. We measure that metric on a subset of images of our proposed PIE-Bench++ dataset where the background can be well defined in that image, e.g., no image style or background editing, and the background is visible after aspect editing. The results are shown in Table 2, where we compare our method with the top performance methods in Table 1. Moreover, we adopt the similar way as calculating the AspAcc-LLaVA to prompt LLaVA [38] for evaluating how the unchanged aspect preserves in the edited image. We also calculate the CLIP [39] score between the target image and the text prompt after removing all edited aspects. The results are reported in Table 2 noted as CLIP and LLaVA, respectively. *Takeaway: preservation is even maintained in ParallelEdits.*

**(c) Branches numbers and aspect grouping.** To demonstrate the effectiveness of our multi-branch design and early aspect grouping, we design additional ablation studies for our method in threefold. (1) We only use one single non-rigid branch to conduct all edits; (2) we remove the aspect categorization process from the pipeline and use the same non-rigid branch for each edit; (3) we adopt one single branch for different type of edits without using any auxillary branches which results a total of three branches (also see Section B for more details). *Takeaway: As shown in Table 3, the multi-branch design and aspect grouping play a significant role in enhancing the performance of our proposed ParallelEdits.*

| | with aspect categorization | with aspect grouping | with auxillary branch | Similarity % | | Aspect Accuracy % | |
|---|---|---|---|---|---|---|---|
| | | | | CLIP↑ | D-CLIP↑ | CLIP↑ | LLaVA ↑ |
| ParallelEdits | × | × | × | 24.32 | 10.45 | 40.97 | 57.67 |
| | × | ✓ | ✓ | 25.14 | 11.97 | 46.66 | 58.37 |
| | ✓ | × | × | 24.50 | 12.33 | 48.08 | 61.22 |
| | ✓ | ✓ | ✓ | **25.70** | **20.70** | **51.05** | **65.19** |

Table 3: **Ablation studies on branch numbers and aspect grouping.**

| | Change | | | | | | | Add | Delete |
|---|---|---|---|---|---|---|---|---|---|
| Asepct Acc-CLIP | Object | Content | Pose | Color | Material | Background | Style | Object | Object |
| P2P [3] | 33.13 | 20.00 | 25.83 | 34.17 | 31.67 | 30.63 | 19.38 | 22.29 | 11.88 |
| MasaCtrl [4] | 40.83 | 23.75 | **40.83** | 20.00 | 30.83 | 26.88 | 29.38 | 37.08 | 28.96 |
| NTI [45] | 48.13 | 41.25 | 23.75 | 51.25 | 24.17 | 51.25 | 22.50 | 40.42 | 32.08 |
| DirectInversion [1] | 40.63 | 26.25 | 23.33 | 40.00 | 25.42 | 32.50 | 25.00 | 30.00 | 20.83 |
| InfEdit [2] | 36.24 | 33.33 | 25.41 | 41.67 | 27.50 | 48.75 | 41.88 | 50.63 | 45.41 |
| PnP [21] | 44.38 | 27.29 | 27.91 | 49.17 | 32.91 | 52.50 | **55.63** | 44.38 | 42.08 |
| ParallelEdits | **51.46** | **44.16** | 39.58 | **60.00** | **47.50** | **60.00** | 50.00 | **56.04** | **52.08** |

Table 4: **Comparison on each category in PIE-Bench++.** Our ParallelEdits achieves the best performance on most of the categories from the dataset.

**(d) Performance comparison on each category.** Recall that our dataset includes nine different categories for editing. We compare the performance of baseline models and our approach across the nine categories, as presented in Table 4. *Takeaway: Our proposed ParallelEdits achieves state-of-the-art performance across most categories.*

**Limitations and Failure Cases.** The proposed ParallelEdits has several limitations. First, it cannot handle the text editing in the image, as shown in the last image pair of Fig. 4. Second, ParallelEdits fails to edit dramatic background changes, as examples shown in the supplementary material.

## 6 Conclusion

In this work, we propose a new research task, multi-aspect text-driven image editing, to modify multiple object types, attributes, and relationships. We introduce a dedicated method, ParallelEdits, to multi-aspect text-driven image editing as an effective and efficient solution to this problem. Due to the lack of evaluation benchmark, we introduce PIE-Bench++, an improved version of PIE-Bench [1] tailored for simultaneous multiple-aspect edits within images. ParallelEdits achieves better quality and performance than existing methods on proposed PIE-Bench++. Our work introduces ParallelEdits, a novel approach that adeptly handles multiple attribute edits simultaneously, preserving the quality of edits across single and multiple attributes through a unique attention grouping mechanism without adding computational complexity. There are several future works we would like to explore. First, different aspects of an image have a specific semantic order. Editing these aspects according to their intrinsic order will simplify the editing process. Secondly, the current ParallelEdits still has limitations, as shown in Fig. 4. It will be of interest to study approaches to improve these aspects.

**Ethics Statement**. In anticipation of contributing to the academic community, we plan to make the dataset and associated code publicly available for research. Nonetheless, we acknowledge the potential for misuse, particularly by those aiming to generate misinformation using our methodology. We will release our code under an open-source license with explicit stipulations to mitigate this risk. These conditions will prohibit the distribution of harmful, offensive, or dehumanizing content or negatively representing individuals, their environments, cultures, religions, and so forth through the use of our model weights.

**Acknowledgement**. This work was supported in part by the National Science Foundation (NSF) Projects under grants SaTC-2153112, No.1822190, and TIP-2137871. Prof. Lokhande thanks support provided by University at Buffalo Startup funds. We thank Sudhir Kumar Yarram for the insightful discussions on the project.

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

# Appendix

## A ParallelEdits: The Algorithm

In this section we provide Algorithm 1: *Early Aspect Grouping* and Algorithm 2: *ParallelEdits on a particular branch*. These algorithms describe the overall idea behind ParallelEdits. They are also pictorially illustrated in Figures 2 and 3 of the main paper. Let us denote an arbitrary branch and the timestep in the diffusion process by $n$ and $t$ respectively. Firstly, in Algorithm 1, we demonstrate how *Early Aspect Grouping* is conducted over the attention maps. Recall that we refer to this as "early" aspect grouping because only a few steps (maximum of 5) are sufficient to perform the grouping. This phase of ParallelEdits takes as an input, the edit action set $\{E^{i \rightarrow j}\}$ and the corresponding cross-attention maps for every token $\mathbf{A}_{src}^j$, and outputs the grouped edit actions set $\bar{\mathcal{A}}_{edt}^c$. Recall from Section 4 of the paper that $E^{i \rightarrow j} \in \{\otimes, \oplus, \ominus, \oslash\}$, with $\otimes$ denoting a swap action, $\oplus$ denoting an add action, $\ominus$ denoting aspect deletion, and $\oslash$ indicating no change in the aspect. Once grouped edit actions set is computed, it is fed into Algorithm 1 to conduct multi-aspect editing and obtain the edited latent features. In Algorithm 2, we implement several operations on the attention masks, similar to the P2P method [3], and describe them as follows.

**Replace**: Swapping token attention mask $\mathcal{M}_{n-1}$ in the prompt from previous branch, overriding $\mathcal{M}_n$;

**Refine**: Injecting only the attention mask that corresponds to the unchanged part of the prompt from $\mathcal{M}_{n-1}$ to $\mathcal{M}_n$;

**Retain**: Keeping the attention mask $\mathcal{M}_n$ unchanged.

---

**Algorithm 1** Early Aspect Grouping

---

**Input:** Edit action set $\{E^{i \rightarrow j}\}$, Cross attention maps $\{\mathcal{M}\}$
1: rigid-edit ← {}, non-rigid-edit ← {}, global-edit ← {}
2: **for** $\mathcal{A}_{edt}^{i \rightarrow j} \in \{E^{i \rightarrow j}\}$ **do**
3:     **if** $\gamma(\bar{\mathcal{M}}_{edt}^j) \geq \beta\gamma(\sum\{\bar{\mathcal{M}}_{edt}\})$ **then**          ▷ This is a global edit
4:         global-edit ← global-edit + $\{E^{i \rightarrow j}\}$
5:     **else if** $\phi(\bar{\mathcal{M}}_{src}^i, \bar{\mathcal{M}}_{edt}^j) < \theta$ **then**          ▷ This is a rigid edit
6:         **for** $\bar{\mathcal{A}}_{edt}^c \in$ rigid-edit **do**
7:             **if** mIoU$(\bar{\mathcal{A}}_{edt}^c, E^{i \rightarrow j} \geq \theta)$ **then**    ▷ $\bar{\mathcal{A}}_{edt}^c$ is a set of grouped edit actions
8:                $\bar{\mathcal{A}}_{edt}^c \leftarrow \bar{\mathcal{A}}_{edt}^c + E^{i \rightarrow j}$
9:             **else**
10:               rigid-edit ← rigid-edit + $E^{i \rightarrow j}$
11:             **end if**
12:         **end for**
13:     **else if** $\phi(\bar{\mathcal{M}}_{src}^i, \bar{\mathcal{M}}_{edt}^j) \geq \theta$ **then**          ▷ This is a non-rigid edit
14:         **for** $\bar{\mathcal{A}}_{edt}^c \in$ non-rigid-edit **do**
15:             **if** mIoU$(\bar{\mathcal{A}}_{edt}^c, E^{i \rightarrow j} \geq \theta)$ **then**
16:                $\bar{\mathcal{A}}_{edt}^c \leftarrow \bar{\mathcal{A}}_{edt}^c + E^{i \rightarrow j}$
17:             **else**
18:               non-rigid-edit ← non-rigid-edit + $E^{i \rightarrow j}$
19:             **end if**
20:         **end for**
21:     **end if**
22: **end for**
**Output:** Grouped edit actions set $\{\bar{\mathcal{A}}_{edt}^c\}$

---

## B Some More Details on ParallelEdits

In the literature [4, 3], image editing processes have been conducted through the implementation of a dual-branch approach. This method involves utilizing a source and target branches for editing.

**Algorithm 2** ParallelEdits on a Particular Branch

---

**Input:** Denoising UNet $\varepsilon_\theta$,
  Grouped edit action $\bar{\mathcal{A}}_{edt}^c$,           $\triangleright$ Output from early aspect grouping
  Latent feature in previous branch and previous timestep $z_{n-1}^t, z_n^{t-1}$,
  Cross attention maps $\{\mathcal{M}\}$,
  Self attention features $Q_{n-1}, K_{n-1}, V_{n-1}$,
  Edit type list: rigid-edit, non-rigid-edit, global-edit
 1: $\mathcal{M}_n \leftarrow \varepsilon_\theta(\bar{\mathcal{A}}_{edt}^c, z_n^{t-1}, t-1)$
 2: **if** $\bar{\mathcal{A}}_{edt}^c \in$ global-edit **then**             $\triangleright$ This is a global edit
 3:   retain($\mathcal{M}_n$)        $\triangleright$ Do not switch attention maps for global edits
 4: **else if** $\bar{\mathcal{A}}_{edt}^c \in$ non-rigid-edit **then**        $\triangleright$ This is a non-rigid edit
 5:   replace($\mathcal{M}_{n-1}, \mathcal{M}_n$)
 6: **else if** $\bar{\mathcal{A}}_{edt}^c \in$ rigid-edit **then**          $\triangleright$ This is a rigid edit
 7:   $\{Q_n, K_n, V_n\} \leftarrow \{Q_n, K_{n-1}, V_{n-1}\}$
 8:   refine($\mathcal{M}_{n-1}, \mathcal{M}_n$)
 9: **end if**
10: $\bar{\mathcal{M}}_n \leftarrow$ binarize($\sum_{m=0}^{m\leq n} \mathcal{M}_m$)
11: $z_n^t \leftarrow \bar{\mathcal{M}}_n \odot z_n^t + (1 - \bar{\mathcal{M}}_n) \odot z_{n-1}^t$
**Output:** Latent feature $z_n^t$

---

Specifically, the source branch is reverted to $z_0$, while the trajectory of the target branch is iteratively adjusted. By computing the distance from the source branch and $\epsilon^{\text{cons}}$ with Latent Consistency Model [32], the target branch is calibrated at each time step.

Our experiments, as seen in Section 5 of the main paper, show the ineffectiveness of a dual-branch procedure for multi-aspect editing tasks. Specifically, a single target branch is inadequate, leading to imperfection in the target image. Thereby we advocate multi-aspect editing through the use of multiple target branches. Each target branch handles a group of aspects, with simpler aspects such as non-rigid local edits directed to initial branches, and more complex aspects such as rigid local edits deferred to subsequent ones. Note that however, all the branches operate simultaneously.

**Auxiliary Rigid / Non-Rigid Branches.** In the main paper, it was noted that there was one dedicated branch for each type of edit: non-rigid, rigid, and global edit. The Early Aspect Grouping algorithm 1 classifies aspects into these three categories. Our experiments revealed that sometimes, due to low overlap between attention maps, aspects may not always be grouped into dedicated rigid or non-rigid branches. In such cases, it becomes necessary to include an auxiliary branch to handle the ungrouped aspects. Therefore, ParallelEdits may involve a single rigid branch and additional auxiliary branches to manage ungrouped aspects, and similarly, a single non-rigid branch and supplementary auxiliary branches to address ungrouped aspects. An ablation study on auxiliary branches is provided in Table 3.

## C More Details on Evaluation Metrics

In this section, we describe more details of our evaluation metrics.

**LLaVA aspect accuracy**. We show how we leverage LLaVA [38] to evaluate the multi-aspect editing accuracy in Fig. 7 and Fig. 8, we also prompt LLaVA [38] for explanation to human readers. LLaVA [38] could provide detailed summary for the image and also explanations for the mismatched between edited aspects and image.

**Other evaluation metrics**. Moreover, even though the Structure Distance [46] has been used in PIE-Bench [1] to evaluate the structure between source and target image while ignoring appearance information, it could not serve as a good evaluation metric for multi-aspect editing. This is due to the fact that the structure of multi-aspect edited target image may necessitate substantial modifications, particularly when it involves adding or removing multiple objects.

Does this image match the elements in []:
"a [sailboat] is silhouetted against the [evening glow] with [snow] mountain in the backgrounds, [winter scene]. "
Exactly follow the return format as a list where 1 is matched and 0 is unmatched, return list only.
The list length is equal to 4.

**LLaVA** [1,1,1,1]

Please provide some explanation.

**LLaVA** The image matches the elements in the description as follows:
1. A sailboat is present in the image, which is silhouetted against the evening glow.
2. The evening glow is visible in the sky, creating a beautiful backdrop for the sailboat.
3. The mountains in the background are covered with snow, which is consistent with the description.
4. The overall scene is a winter scene, as indicated by the snow-covered mountains and the calm, cold appearance of the water.
Therefore, all elements in the description are matched by the image, resulting in a list of 4 ones.

Does this image match the elements in []:
"a man [standing] in a boat is silhouetted against the sunset and [ducks] on the water with [Alps mountain] in the background"
Exactly follow the return format as a list where 1 is matched and 0 is unmatched, return list only.
The list length is equal to 4.

**LLaVA** [1,1,1,1]

Please provide some explanation.

**LLaVA** The image matches all the elements in the given description:

1. A man is standing in a boat.
2. The man is silhouetted against the sunset.
3. There are ducks on the water.
4. The background features a mountain range, which could be described as Alps-like.
Therefore, the elements in the description are all present in the image, and the list would be [1, 1, 1, 1].

Does this image match the elements in []:
"a [boat] in a lake with a [house] and [ice mountain] in the background"
Exactly follow the return format as a list where 1 is matched and 0 is unmatched, return list only.
The list length is equal to 3.

**LLaVA** [1,1,1]

Please provide some explanation.

**LLaVA** The image shows a boat on a lake with a house and a mountain in the background. The mountain appears to be covered with snow, which could be described as an "ice mountain." The elements in the brackets match the description provided:

A boat is present in the image.
A house is visible in the background.
The mountain in the background is covered with snow, which could be described as an "ice mountain."

Therefore, the elements in the brackets match the description, and the list would be [1, 1, 1].

Figure 7: **Examples of prompting LLaVA for aspect accuracy measurement in cases of successful editing.**
LLaVA can effectively illustrate if and how the target image and edits are misaligned.

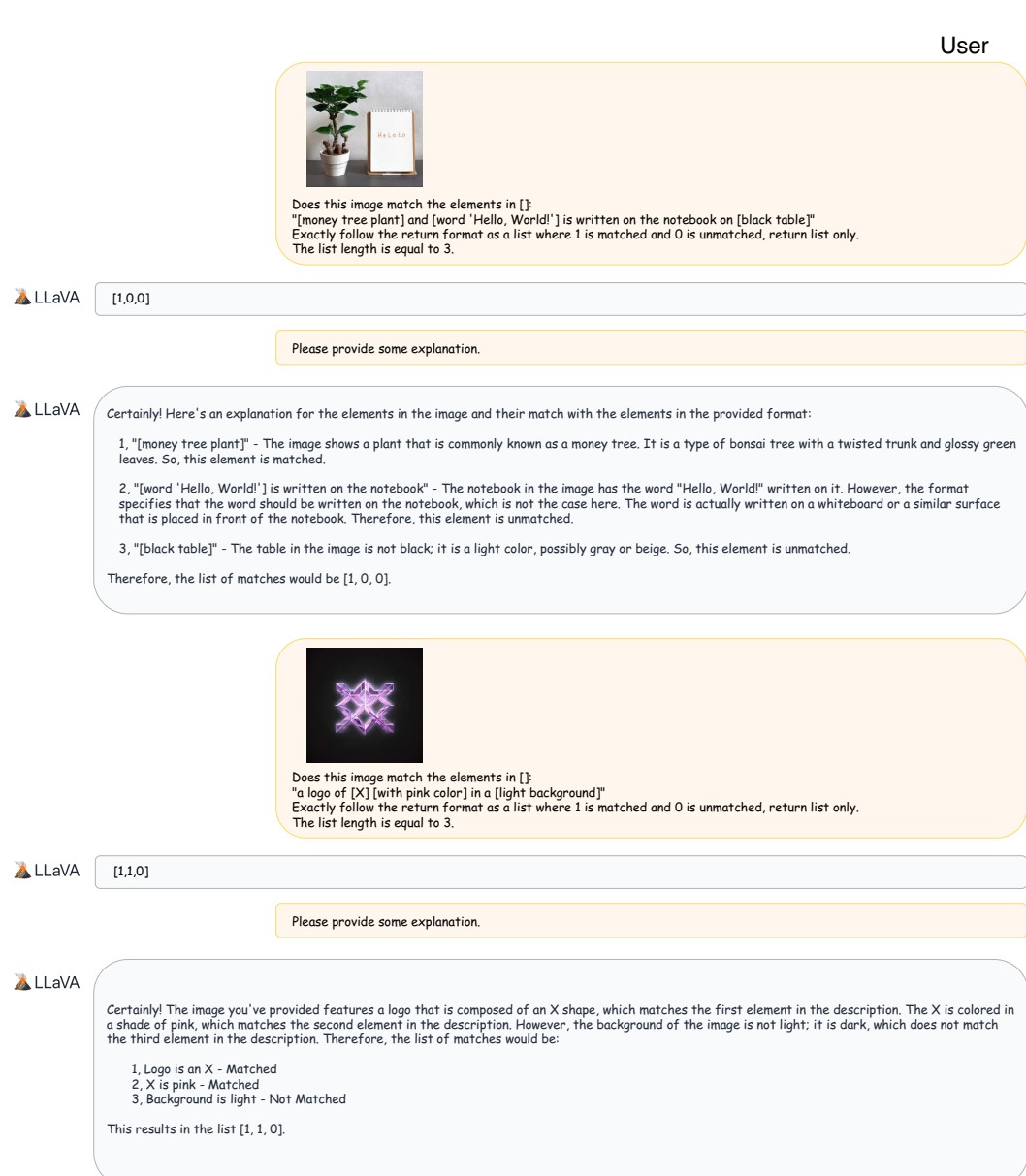

Figure 8: **Examples of prompting LLaVA for aspect accuracy measurement in cases of unsuccessful editing.**

## D    Implementation Details

Our proposed ParallelEdits is based on the Latent Consistency Model [32], with the publicly available LCM [†] which is finetuned from Stable Diffusion v1.5. We then follow [2] to leverage their proposed inversion-free technique in ParallelEdits for image editing. During sampling, we perform LCM sampling [32] with 15 denoising steps, and the classifier-free guidance (CFG) is set to 4.0. ParallelEdits can control the editing strength by adjusting the CFG . There's a trade-off between achieving satisfactory inversion and robust editing ability. A higher CFG tends to produce stronger editing effects but may lower inversion results and identity preservation. We also set the hyper-parameter $\theta$ as 0.9 and $\beta$ as 0.8 in our experiments, where $\theta, \beta$ are used to determine the edit type of a given edit action.

---

[†] https://huggingface.co/SimianLuo/LCMDreamshaperv7

|  | Change | | | | | | | Add | Delete |
|  | Object | Content | Pose | Color | Material | Background | Style | Object | Object |
|---|---|---|---|---|---|---|---|---|---|
| #Edited Aspect | 302 | 98 | 120 | 188 | 99 | 112 | 165 | 178 | 119 |
| #Edited Token | 316 | 155 | 227 | 205 | 116 | 175 | 424 | 507 | 381 |

Table 5: **Summary of Editing Types and Categories in PIE-Bench++ dataset.** There are 10 different categories in PIE-Bench++ and a total number of 700 images.

In the inversion-free multi-branch editing approach, for $1 < n < N$, the noise estimation is also conditioned on a text conditioning $c_n$ in branch $n$. This can be expressed as $\epsilon(n)_\tau^{\text{edt}} = \epsilon_\theta(z(n)_\tau^{\text{edt}}, \tau, c_n)$. Here, $c_1$ corresponds to the source prompt, $c_N$ corresponds to the target prompt, and $c_n$ represents the prompt that includes all aspect edits up to branch $n$.

# E    Additional Details of PIE-Bench++

## E.1    PIE-Bench++ Details

Unlike existing benchmarks that primarily focus on single-aspect edits, PIE-Bench++ is tailored to multiple aspect edits, reflecting the complexities inherent in real-world editing tasks. Our enhanced dataset, PIE-Bench++, builds upon the PIE-Bench [1] by incorporating 700 images across nine diverse categories, covering both natural and artificial scenes, with a significant focus on multi-aspect editing scenarios. Specifically, the Change Object category involves swapping objects in the scene with different yet reasonable alternatives. Add Object adds new elements to the scene. Delete Object focuses on removing objects, testing the model's ability to erase elements seamlessly. Change Object Content alters the content of specific objects, such as changing the design on a shirt or the pattern on a wall. Change Object Pose includes changes in the shape of objects, humans, or animals. Change Object Color assesses the model's ability to apply accurate color changes. Change Object Material evaluates the rendering of different textures and materials. Change Background involves editing scenarios where there is a distinct foreground object and a main background. This type of edit focuses on seamlessly integrating new background elements while preserving the integrity of the foreground object. Change Image Style involves the application of style transfer techniques to the entire image while ensuring the original content remains intact. For example, this could involve transforming a photograph to adopt a cartoon style. Each category is carefully curated to provide a comprehensive evaluation of the dataset's multi-aspect editing capabilities, the summary of the dataset is shown in Table 5.

## E.2    Dataset Annotation

The annotation process involves a primary annotator who labels the source prompt, describing the original image, and the target prompt, which outlines the desired modifications to generate the target image. The target prompt is carefully annotated to include all editing pairs expected to be reflected in the target image. Subsequently, a second annotator reviews the annotations for accuracy and consistency, ensuring the reliability of the dataset. The majority of target prompts in PIE-Bench++ feature at least two edited aspects. Nevertheless, within the categories that solely changing background and image styles, the number of edits is usually constrained to one or two aspects. This limitation is due to the intrinsic characteristics of these attributes, such as each image having only one background or style.

**Annotation format details**. Each image in the dataset annotation is associated with key elements as shown in Fig. 9: a source prompt, a target prompt, an edit action, and a mapping of aspects. The edit action specifies the position index in the source prompt where changes are to be made, the type of edit to be applied, and the operation required to achieve the desired outcome. The aspect mapping connects objects undergoing editing to their respective modified attributes, enabling the identification of which objects are subject to editing.

| Source Image | Text-based annotation |
|---|---|
|  | "source_prompt": "a colorful bird standing on a branch",

"target_prompt": "a brown owl standing on a red flower",

"edit_action": {"owl":{"position":2,"edit_type":1,"action":"bird"},
    "brown":{"position":1,"edit_type":6,"action":"colorful"},
    "flower":{"position":6,"edit_type":1,"action":"branch"},
    "red":{"position":6,"edit_type":6,"action":"+"}},

"aspect_mapping": {"owl":["brown"],"flower":["red"]} |
|  | "source_prompt": "the galaxy over the durdle door",

"target_prompt": "the pink sunset and rainbow over the durdle door",

"edit_action": {"pink":{"position":1,"edit_type":6,"action":"+"},
    "sunset":{"position":1,"edit_type":8,"action":"galaxy"},
    "and rainbow":{"position":2,"edit_type":2,"action":"+"}},

"aspect_mapping": {"sunset":["pink"],"rainbow":[]} |
|  | "source_prompt": "a slanted mountain bicycle on the road in front of a building",

"target_prompt": "a slanted rusty mountain motorcycle on the road in front of a fence",

"edit_action": {"rusty":{"position":2,"edit_type":7,"action":"+"},
    "motorcycle":{"position":3,"edit_type":1,"action":"bicycle"},
    "fence":{"position":11,"edit_type":8,"action":"building"},
    "on the road":{"position":4,"edit_type":3,"action":"-"}},

"aspect_mapping": {"motorcycle":["rusty"],"fence":["red"],"road":[]} |
|  | "source_prompt": "a round cake with orange frosting on a wooden plate",

"target_prompt": "a square cake with strawberry frosting on a plastic plate",

"edit_action": {"square":{"position":1,"edit_type":4,"action":"bird"},
    "strawberry frosting":{"position":4,"edit_type":6,"action":"orange frosting"},
    "plastic":{"position":8,"edit_type":7,"action":"wooden"}},

"aspect_mapping": {"cake":["square","strawberry frosting"], "plate":["plastic"]} |

Figure 9: **Annotation examples from PIE-Bench++**. Each annotation containing a Source Prompt, Target Prompt, Edit Action, and Aspect Mapping. Edit action contains the specific instructions including the desired modification index in source prompt as position, edit type among 9 catergories and the action $\in \{\otimes, \oplus, \ominus\}$. The aspect mapping indicts the pair between object and attribute.

# F   Additional Qualitative Results

We also provide more qualitative results in Fig. 10, showing the effectiveness of our proposed method in handling multi-aspect editing tasks. These examples showcase the model's proficiency in executing intricate edits. For instance, as depicted in Fig. 10 (b), our method successfully removes a cup while accurately reconstructing the obscured parts of the lamp behind it. In Fig. 10 (a), the model demonstrates its ability to swap and add aspects, while preserving the composition of the scene. The results underscore the model's adeptness in interpreting and executing sophisticated editing instructions, leading to visually consistent and contextually fitting edited images. Additional, we also provide the results for sequential editing methods with different editing order in Fig. 11 and Fig. 12.

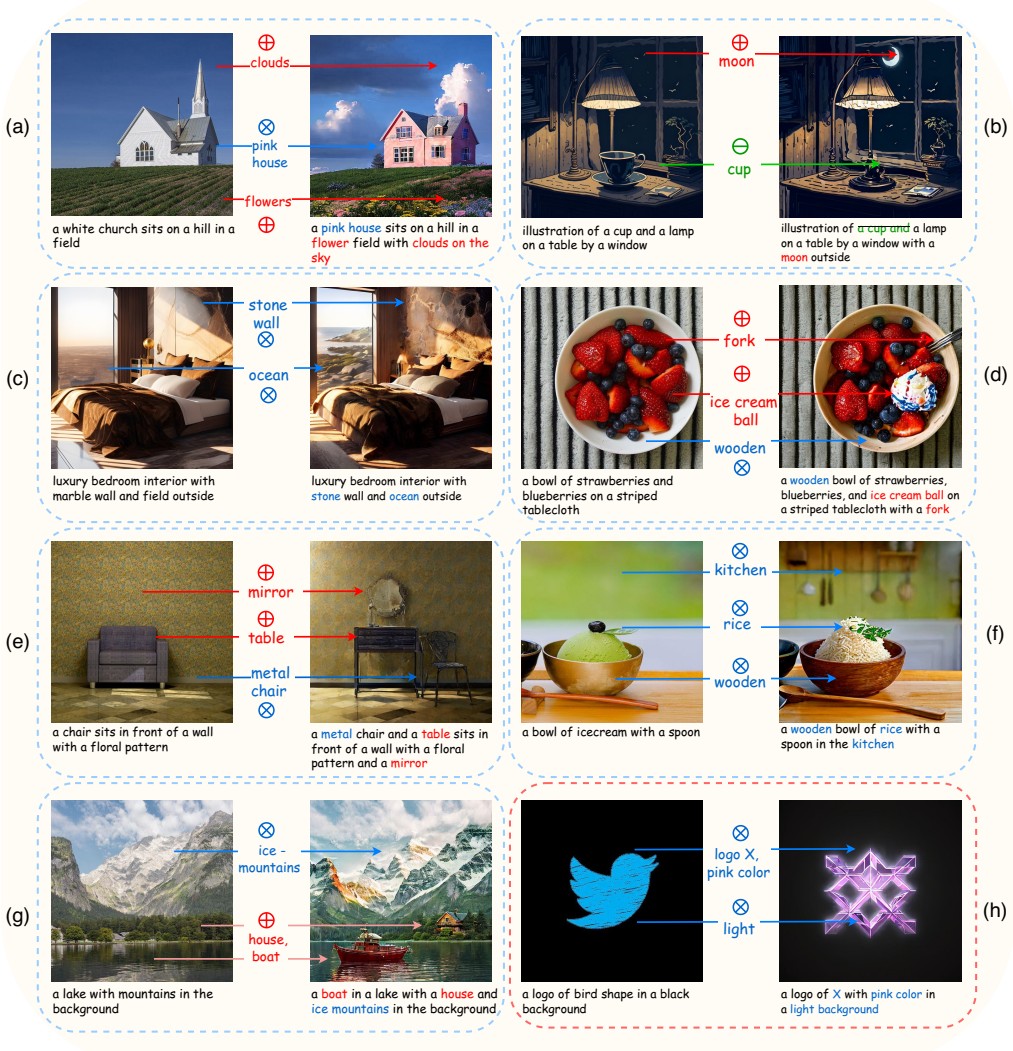

Figure 10: **Qualitative results from ParallelEdits.** ParallelEdits is able to swap, add and delete multiple aspects. The last image pair is a failure case of ParallelEdits.

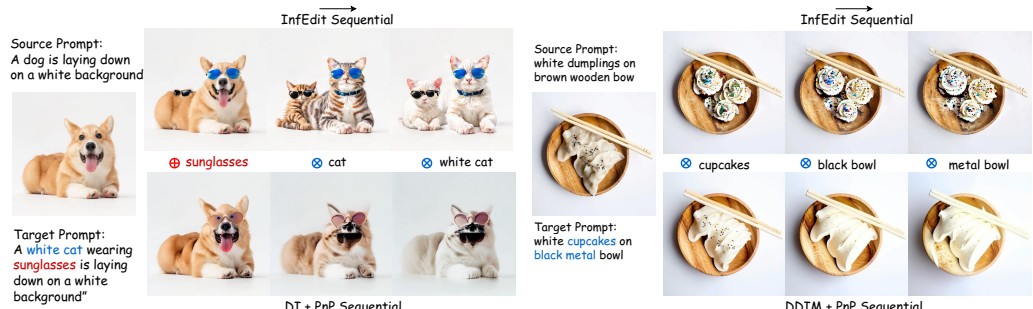

Figure 11: **Sequential editing using single-aspect text-driven image editing methods.** The sequential editing might accumulate errors and undo previous edits. It also fails to edit significantly overlapped objects.

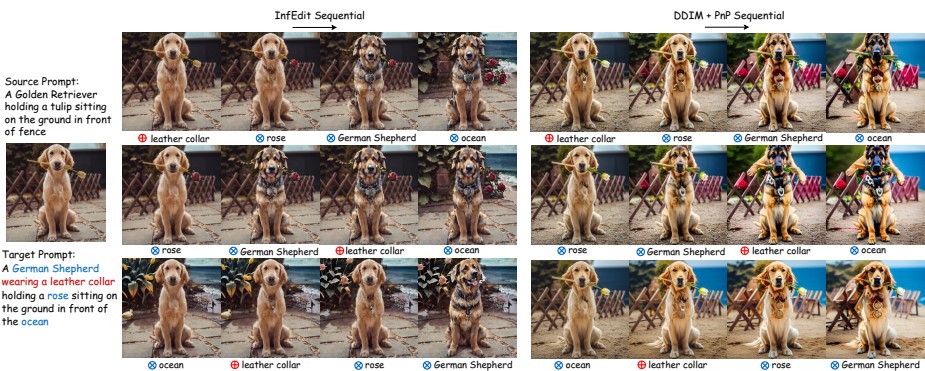

Figure 12: **Sequential editing with different orders.** Sequential editing with different orders can yield varying final results. Additionally, it may lead to error accumulation and potentially overwrite previous edits.

