# OpenReview forum: "ParallelEdits: Efficient Multi-Aspect Text-Driven Image Editing with Attention Grouping"
_NeurIPS.cc/2024/Conference — NeurIPS 2024 poster_

### Official Review · Reviewer_8ae7 · 2024-06-23

**Soundness:** 3
**Presentation:** 2
**Contribution:** 2
**Rating:** 6
**Confidence:** 5

**Summary:**

This paper aims to make multiple objects or attributes editing, while preserving the quality, named ParallelEdits. The paper also introduces a new dataset PIE-Bench++. Experimental results on both PIE-Bench and PIE-Bench++ demonstrate that this method outperforms many existing editing techniques.

**Strengths:**

Simultaneously editing multiple objects or attributes is interesting.

The finding that the order in which aspects are modified when applying single-aspect text-driven image editing methods can affect the quality is interesting.

**Weaknesses:**

1.	Direction Inversion is solely a method for inverting real images, not for editing them. Figure 1 is confusing in this context. Direction Inversion can be combined with various editing methods like P2P, PnP, P2P-zero, and MasaCtrl. Which editing method is Direction Inversion combined with in the third row of results? Why is this particular editing method shown?
2.	Same problem in the description in the introduction (Line 22, latest methods [3, 1, 4]; Line36, Unlike traditional editing methods (e.g., [1, 2])). [1] (Direction Inversion) is not a editing method.
3.	The paper claims that ParalleEdits can edit multiple objects or attributes simultaneously. Figure 1. illustrates the swapping of multiple objects within the same image, but it only demonstrates the addition of a single object. Can ParalleEdits add multiple objects within a single image, such as both a necktie and sunglasses to a cat?
4.	In aspect grouping, the paper does not explain how to get the attention map M for a real image.
5.	In Figure 2, why does the attention map from the original image combine with the target prompt.
6.	This paper divides aspect grouping based on cross-attention. What if cross-attention is inaccurate? For example, in Figure 2, will each image clearly show a "ducks" cross-attention map when "ducks" are added to the image?
7.	In Table 1, P2P is an editing method for generated images. When applying it to real images (PIE-Bench++ dataset), it needs to be combined with an inversion technique (i.e., NTI, DI). Need to explain.
8.	The finding that the order in which aspects are modified when applying single-aspect text-driven image editing methods can affect the quality is interesting. However, the paper does not conduct qualitative and quantitative experiments to support this claim.

Further comments:
+ Line 119, should z be z_0 ?
+ Line 135, Xu et al. [3] seems not to use \cite{} to link the reference?
+ Line148~Line149, the definition of tokens is not clear for me. Do the tokens represent all nouns and adjectives in a text prompt?

**Questions:**

Does StyleDiffusion (StyleD) refer to “Controllable disentangled style transfer via diffusion models” or “Prompt-Embedding Inversion for Text-Based Editing” ? If it is the former, please explain why this paper is compared with style transfer methods. If it is the latter, please correct it.

Simultaneously editing multiple objects or attributes is interesting. But some aspects of the method are not adequately explained (e.g., how to get the attention-map M), and some references within the paper are quite casual (e.g., DirectInversion and StyleDiffusion).

The weaknesses low my initial rate.

---
**Update after author responses**
Thanks to the authors for their efforts. I'm happy to see the extended discussion and additional comparisons that have resolved my concerns.

**Limitations:**

Yes

---

> ### Author Rebuttal · Authors · 2024-08-07
>
> **Q: Clarification of the inversion and editing method.**
>
> **A:** We agree with the reviewer. Each baseline in Table 1 and throughout the paper includes a submethod for inversion and another for editing. The table below details these submethods for each baseline. We will update line 36 of the introduction, Table 1/Figure 5 captions, and lines 269-272 to specify the submethods used for inversion and editing for each baseline.
>
> |Method stated in paper|MasaCtrl|P2P|NTI|  StyleD | PnP      |DI|
> |-|-|-|-|-|-|-|
> | Inversion technique | DDIM | DDIM | NTI |StyleD|DDIM|DI|
> |Editing technique |MasaCtrl|P2P|P2P|P2P|PnP|PnP|
>
> **Q:** P2P is an editing method for generated images and needs inversion method to edit real image.
>
> **A:** For P2P, we adopt the default settings in the original P2P paper [1] (introduced in the Real Image Editing section) to conduct DDIM inversion to edit real images.
>
> **Q:** Is ParalleEdits able to add multiple object in the image?
>
> **A:** ParallelEdits can add multiple objects to an image, as shown in Fig 4 of the main paper (adding harness and flowers) and Fig 10 in the Appendix (d,e,f). PIE-Bench++ includes test samples for evaluating multiple object addition. We tested the example provided by the reviewer—a cat wearing a necktie and sunglasses—as detailed in the document attached in the general response (Figure 1), and we welcome any additional challenging examples the reviewer wishes to suggest.
>
> **Q:** Need to explain how to get the attention map for a real image.
>
> **A:** An attention map is generated by running a few diffusion steps (up to 5 steps) as in line 472 of the main paper and it is created by averaging cross-attention matrices of UNet layers, as outlined in P2P (see section 3.2 of [1]). The main paper (line 175) mentions that the aspect grouping process uses a dual-branch diffusion model conditioned on source and target prompt to generate and group attention map. Attention maps from the two branches are associated with source prompt tokens and target prompt tokens, and the attention maps of source branch correspond to the aspects in real/source images. We will include these details in the appendix of the revised manuscript.
>
> **Q:** Figure 2, Why does the attention map from the original image combine with the target prompt.
>
> **A:**  The source and target attention maps are associated with source and target prompt, while both source and target attention maps have been to used to identify the editing type, as shown in Figure 3. We visualize the target attention map overlaid in the original image to illustrate which part of the original image need to be edited based on the attention map.
>
> **Q:** What if cross-attention is inaccurate? For example, is the duck attention map clear to show?
>
> **A:** Yes, adding "ducks" to an image creates a cross-attention map as Figure 3 of the main paper shown. Current image editing works [1,2,3,4] have also used attention map to indict object layouts. The iterative diffusion process is able to generate an accurate object layouts in the attention maps ( as described in section 3.2 of P2P paper [1]). Moreover, attention maps have also been utilized for even more intricate tasks such as image segmentation [5]. Although failure situations resulting from faulty attention maps do occur, they are rare.
>
> **Q:** Need qualitative and quantitative experiments to support the claim that order in which aspects are modified when applying single-aspect text-driven image editing methods can affect the quality.
>
> **A:**. Thanks for pointing this out. We have conducted additional qualitative and quantitative experiments to support the claim, please check Table 2 and Figure 4 attached in the general response. We found that the AspAcc-CLIP is significantly different when using varying orders for each single-aspect editing method, and each method achieves the best performance with different orders. Hence, it is hard to choose a proper editing order for sequential editing, not to mention that sequential editing is extremely slow.
>
> **Other minor issues**
>
> **Q:** z should be $z_0$ in Line 119.
>
> **A:** Yes, we will reviese the manuscript.
>
> **Q:** The definition of tokens is not clear.
>
> **A:** The tokens are words obtained from the tokenizer which includes nouns, adjectives and other words splitted by spaces. For example, in the figure 3 the prompt has been broken down into tokens, all the words are token but only the underlining ones are asepcts.
>
> **Q:** Misused of reference.
>
> **A:** Thanks for pointing this out, StyleDiffusion refers "Prompt-Embedding Inversion for Text-Based Editing", and Xu et al. refers the [3] Inversion-Free Image Editing with Natural Language. We will update the references in the revision.
>
> [1] Prompt-to-Prompt Image Editing with Cross Attention Control, Hertz et al, ICLR23'
>
> [2] DiffEdit: Diffusion-based semantic image editing with mask guidance, Couairon et al, ICLR 23'
>
> [3] Inversion-Free Image Editing with Natural Language, Xu et al, CVPR24'
>
> [4] MasaCtrl: Tuning-Free Mutual Self-Attention Control for Consistent Image Synthesis and Editing, Cao et al, ICCV23'
>
> [5] Open-Vocabulary Attention Maps with Token Optimization for Semantic Segmentation in Diffusion Models, Marcos et al, CVPR24'

---

> > ### Comment · Reviewer_8ae7 · 2024-08-09
> > **Response**
> >
> > I would like to thank the authors for the extensive and detailed rebuttal, it has been very helpful. There remain some questions that would be helpful of this paper.
> >
> > **Q2. Applying DDIM Inversion for real image inversion.**
> >
> > In the P2P paper, the authors found that the inversion is not sufficiently accurate in many cases, or that using a mask yields a more accurate inversion. The authors should specify which method was used and provide detailed explanations in the paper. Real image inversion and editing represent only a small portion of the P2P paper, and thus, are not the primary focus of P2P. Comparing the inversion method in P2P with methods specifically designed for real image inversion is not fair, as P2P typically combines other inversion methods when editing real images.
> >
> > In ParallelEdits, which CFG is used in DDIM inversion? Using a CFG of 1 typically results in satisfactory inversion of a real image but may reduce editing ability. Conversely, a CFG of 7.5 often leads to reconstruction failure.
> >
> > ---
> >
> > I would like to thank the authors for their response. With the additional results and clarifications provided, I am pleased to increase my rating.

---

> > > ### Author Response · Authors · 2024-08-10
> > > **Additional results and clarifications**
> > >
> > > We are glad that the reviewer finds our rebuttal helpful. The reviewer's suggestions have been valuable, and we welcome any further input.
> > >
> > > ***Q1: The authors should specify which method was used, DDIM inversion or naive DDPM inversion with mask.***
> > >
> > > ***A1:*** Our paper employs DDIM for inversion and P2P for editing in the P2P baseline, a combination that does NOT require masks, aligning with findings from previous research [1,2]. Following the reviewer's suggestions, we will include this discussion about mask-guided inversion into the paper. Note that, mask guidance uses a naive DDPM inversion by simply adding noise then denoising. However, mask-guided inversion P2P+DDPM cannot be used for non-rigid edits (pose/layout change) and global edits (style changes), and does not ensure identity preservation, often resulting in significant changes to the image content (please refer to the Introduction and Related Work Sections of [4] and Related work Section of [3]).
> > >
> > >
> > > ***Q2: Comparing the inversion method in P2P with methods specifically designed for real image inversion is not fair, as P2P typically combines other inversion methods when editing real images.***
> > >
> > > ***A2:*** Comparing the inversion method in P2P with others is only a part of our overall analysis. We have compared with P2P with other real image inversion method like StyleD and NTI, also other editing techniques such as PnP, MasaCtrl, and InfEdit. Regardless of the inversion or editing method of choice, we observe that ParallelEdits consistently outperforms benchmarks in aspect accuracy, aspect preservation, and CLIP/D-CLIP scores. Although we could explore more detailed comparisons across various baselines, it is important to note that current prevailing techniques in the research community primarily focus on single-aspect editing. Our research highlights the necessity to transition towards multi-aspect editing.
> > >
> > >
> > > ***Q3: In ParallelEdits, which CFG is used in DDIM inversion? Using a CFG of 1 typically results in satisfactory inversion of a real image but may reduce editing ability. Conversely, a CFG of 7.5 often leads to reconstruction failure.***
> > >
> > > ***A3:*** Great question! Adjusting the CFG (Classifier-Free Guidance) can influence editing quality and image/identity preservation. In our experiments, we used a moderate CFG setting of 4.0 to balance aspect accuracy, preservation metrics, and editing quality. For more information on hyper-parameters, please refer to Section E, Implementation Details, in the Appendix.
> > >
> > > ---
> > > [1] Inversion-Free Image Editing with Natural Language, Xu et al, CVPR24'
> > >
> > > [2] Boosting Diffusion-based Editing with 3 Lines of Code, Ju et al, ICLR24'
> > >
> > > [3] An Image is Worth One Word: Personalizing Text-to-Image Generation using Textual Inversion, Gal et al, ICLR23'
> > >
> > > [4] MasaCtrl: Tuning-Free Mutual Self-Attention Control for Consistent Image Synthesis and Editing, Cao et al, ICCV23'

---

> > > > ### Comment · Reviewer_8ae7 · 2024-12-19
> > > > **The authors did not fix the issue they promised to address in the Camera-Ready version.**
> > > >
> > > > The authors acknowledge the issue and have promised to correct the reference. However, this correction was not made in the Camera-Ready version. Please ensure that the correction is implemented..
> > > >
> > > > ---
> > > > Q: Misused of reference.
> > > >
> > > > A: Thanks for pointing this out, StyleDiffusion refers "Prompt-Embedding Inversion for Text-Based Editing", and Xu et al. refers the [3] Inversion-Free Image Editing with Natural Language. We will update the references in the revision.

---

> > > > > ### Public Comment · ~Mingzhen_Huang1 · 2024-12-20
> > > > >
> > > > > The authors thank the reviewer for the reminder, the reference has been corrected now.

---

### Official Review · Reviewer_J7sk · 2024-07-08

**Soundness:** 3
**Presentation:** 2
**Contribution:** 3
**Rating:** 5
**Confidence:** 5

**Summary:**

The paper introduces ParallelEdits, a method that manages simultaneous edits efficiently without compromising quality. This is achieved through a novel attention distribution mechanism and multi-branch design. Additionally, the authors present the PIE-Bench++ dataset, an expanded benchmark for evaluating multi-object and multi-attribute image-editing tasks.

**Strengths:**

1. The task of multi-aspect editing seems interesting and useful in real life.
2. The proposed method is efficient for application.
3. The overall editing results seem promising.

**Weaknesses:**

1. The authors claim the proposed method is based on the DDCM process [1]. However, when I checked the source paper of [1], I found something strange, especially its Algorithm 1. The dozens of iterations come to a simple conclusion that the output $z$ actually equals $z_0$, which means there is no need for such Virtual Inversion. Also, this process cannot be interpreted by Consistency Models. Hence, I wish the authors could revise their theoretical bases.

2. I think it is better to present the intermediate editing results for sequential editing, which can help readers identify what type of editing the previous methods are not good at.

3. I noticed that there was a benchmark [2] proposed for image editing. I advise the authors to use it for comprehensive evaluation.

[1] Inversion-Free Image Editing with Natural Language (https://arxiv.org/abs/2312.04965)

[2] Diffusion-Model Based Image Editing: A Survey (https://arxiv.org/abs/2402.17525)

**Questions:**

I noticed that the man has changed after editing in Fig. 2. Could you please explain and address it?

**Limitations:**

See above.

---

> ### Author Rebuttal · Authors · 2024-08-07
>
> **Q:** The DDCM process in [1] comes to a simple conclusion that the output z equals $z_0$ after dozens of iterations.
>
> **A:** Indeed, ParallelEdits' source branch uses DDCM, as described in [1], which uses the consistency sampling step popular in consistency model literature [6,7]. DDCM process for source branch was chosen based on [1]'s observations that the method does not suffer cumulative errors over sampling, making the overall method more efficient (fewer samples to generate target image), which is desirable for image-editing applications. Although $z=z_0$ in every iteration of Algorithm 1, the difference between the reconstructed output $z_\tau$ and $z_0$ as captured by $\epsilon_\text{cons}$ continues to evolve (see Figure 3b of [1]). The target branch is calibrated using $\epsilon_\text{cons}$ thereby introducing the desired edit.
>
> **Q:** Results about intermediate editing results for sequential editing
>
> **A:** We have included further results in Figures 3 and 4 of the attached document in the general response. As previously stated in line 48 of the main paper, sequential editing is an ineffective approach due to the potential for undoing previous edits and accumulating errors over time.
>
> **Q:** Additional benchmark needs to be included for comprehensive evaluation.
>
> **A:** We thank the reviewer for highlighting this interesting benchmark. Although it predominantly contains single-aspect editing samples, unlike the more complex multi-aspect samples in PIE-Bench++, we evaluated our method on it and reported the results in Table 1 of the global rebuttal. Our method achieves state-of-the-art performance in terms of the overall LMM score proposed in [2]. Inspired by this, we plan to release a benchmark focused on multi-aspect editing, using PIE-Bench++ and the Aspect-accuracy/preservation metrics from our paper. We are confident it will gain strong recognition from the community.
>
> **Q:** Explanation of man has changed after editing in Fig. 2.
>
> **A:** We appreciate the reviewer’s subtle observation. As shown in Figure 2, the “boat” object nearby has been transformed to “white rubber boat” and leaks into the “man” attention map, and the man has been edited to fit the context of changing a traditional boat to a modern rubber boat. Similar with other attention-map-based approaches [1,3,4,5], the attention maps could leak, even though such minor changes do not affect the overall context of this figure. Finally, such leakage can be easily solved by applying a higher threshold to the attention map.
>
> [1] Inversion-Free Image Editing with Natural Language, Xu et al, CVPR24'
>
> [2] Diffusion-Model Based Image Editing: A Survey, Huang et al, ArXiv24'
>
> [3] Prompt-to-Prompt Image Editing with Cross Attention Control, Hertz et al, ICLR23'
>
> [4] DiffEdit: Diffusion-based semantic image editing with mask guidance, Couairon et al, ICLR 23'
>
> [5] MasaCtrl: Tuning-Free Mutual Self-Attention Control for Consistent Image Synthesis and Editing, Cao et al, ICCV23'
>
> [6] Latent consistency models: Synthesizing high-resolution images with few-step inference, Luo et al, ArXiv23'
>
> [7] Consistency Models, Song et al, ICLR23'

---

> > ### Author Response · Authors · 2024-08-14
> > **Thanks for the feedback**
> >
> > We thank Reviewer J7sk for the positive and constructive feedback. As the rebuttal period is ending, we wanted to check if there are any remaining questions. We have provided additional intermediate results on the baseline method (sequential editing) as requested and evaluated our approach on a different benchmark, where our method still outperforms the state-of-the-art. We appreciate the suggestion and will release PIE-Bench++ as a benchmark for multi-aspect editing. We are glad to address any further questions or suggestions from the reviewer that could help enhance our manuscript.

---

### Official Review · Reviewer_6th7 · 2024-07-12

**Soundness:** 3
**Presentation:** 3
**Contribution:** 3
**Rating:** 6
**Confidence:** 3

**Summary:**

In this paper, the authors present a novel multi-aspect image editing method ParallelEdits, by incorporating the attention distribution mechanism and multi-branch editing. Besides, this paper introduces a new dataset PIE-Bench++ for evaluating multi-aspect image editing. Extensive experiments demonstrate the effectiveness of this method.

**Strengths:**

1. The paper is well-written and well-organized.
2. The proposed method is novel and effectively addresses the multi-aspect image editing problem.
3. The results appear to have impressive visual effects. Edits of different aspects are combined seamlessly.
4. The authors conducted concrete experiments, providing rich quantitative results and qualitative results to demonstrate the effectiveness of the proposed ParallelEdits.

**Weaknesses:**

1. The pairing process of $E^{i \rightarrow j}$ imposes an additional burden on users, especially when multiple objects are present in the source image.
2.  It is not clear how to choose the hyperparameter $\lambda$ for assigning a type for rigid and non-rigid edit actions, and how it affects the cross-branch interaction and editing results.
3. This paper lacks a discussion on controlling the editing strength of multiple editing actions, which is crucial for harmonic and flexible editing.

**Questions:**

See the weakness part above.

**Limitations:**

The author has discussed the limitations and potential negative societal impacts.

---

> ### Author Rebuttal · Authors · 2024-08-07
>
> **Q:** The pairing process imposes an additional burden on users when performing multi-object editing.
>
> **A:** The algorithm receives the pairing process (editing action) to determine which aspect is added, removed, swapped, or left unaltered. Meta-data of this form is not a burden but a necessity because it precisely reflects the user's intent when editing without errors. A user-friendly UI can conveniently capture such meta-data. Moreover, prior works also benefit from meta-data e.g. [1] needs rich text input and [2,3] need editing pairs.
>
> **Q:** How the hyperparameter $\lambda$ to be chosen and how it affects the editing.
>
> **A:** The hyperparameter $\lambda$ is the threshold for determining whether an edit is classified as a rigid edit or a non-rigid edit (see equation 3 of the paper). The hyperparameter $\lambda$ was selected through cross-validation by identifying the optimal $\lambda$ that gave the highest aspect accuracy. Experiments showed that both quantitative results and qualitative results do not vary for different choices of $\lambda$. Please see the table below and the Figure 2 of the PDF appended to this rebuttal. We will gladly share additional results if the reviewer desires.
>
>
> |    $\lambda$    | 0.8   | 0.85  | 0.9 (ours)   | 0.95  |
> |--------|-------|-------|-------|-------|
> | AspAcc-CLIP | 50.09 | 50.68 | 51.05 | 50.53 |
>
> **Q:** Lacking discussions on controlling the editing strength of multiple editing.
>
> **A:** We concur with the reviewer that edit strength control is important. As shown in [4,5], single attribute editing tasks provide clear editing strength and flexibility. As for multi-aspect editing, the reviewer would agree that it is unclear if edit for multiple aspect is itself possible. Moreover, evaluation metrics and benchmark datasets for multi-aspect editing are not yet community-standardized. Thus, instead of handling editing flexibility, contemporary efforts [6,7] focus on feasibility aspects of simultaneous multiple edits.
>
>
> [1] Expressive text-to-image generation with rich text. Ge et al, ICCV23.
>
> [2] Prompt-to-prompt image editing with cross attention control. Hertz, et al, ICLR23.
>
> [3] Null-text inversion for editing real images using guided diffusion models. Mokady, et al, CVPR23.
>
> [4] An edit friendly ddpm noise space: Inversion and manipulations. Huberman-Spiegelglas et al, CVPR24.
>
> [5] Imagen editor and editbench: Advancing and evaluating text-guided image inpainting. Wang, et al, CVPR23.
>
> [6] Ground-A-Score: Scaling Up the Score Distillation for Multi-Attribute Editing. Chang, et al, Arxiv24.
>
> [7] Iterative Multi-granular Image Editing using Diffusion Models. Joseph, et al, WACV24.

---

> > ### Comment · Reviewer_6th7 · 2024-08-09
> >
> > Thanks for authors' rebuttal. There remain some questions regarding the paper and rebuttal.
> >
> > - How did you collect the pairing information for PIE-Bench++?
> > - The hyperparameter $\lambda$ is the threshold for determining whether an edit is classified as a rigid edit or a non-rigid edit. Could you further explain which edits are rigid edit and which are non-rigid in Fig.2 of the PDF appended and how they change the outputs in Fig.2?
> > - Can you control editing strength for ParallelEdits? If yes, how do you control it? If no, does it mean the editing result for a given pair prompt is fixed?

---

> > > ### Author Response · Authors · 2024-08-10
> > > **Additional clarifications**
> > >
> > > We thank the reviewer for additional questions and helpful feedback.
> > >
> > > ***Q1: How to collect the pairing information for PIE-Bench++?***
> > >
> > > ***A1:*** All editing pairs have been manually annotated to establish editing correspondence like PIE-Bench. To build a robust dataset for community use, annotators manually labeled 700 image prompts.
> > >
> > > ***Q2: Need further explain which edits are rigid edit and which are non-rigid in Fig.2 of the PDF appended and how they change the outputs in Fig.2?***
> > >
> > > ***A2:*** In Fig. 2 of the rebuttal PDF, the "harness" is always non-rigid, whereas the "horse" and "field" are rigid edits. According to line 194 of the main paper, edits with an overlap greater than $\lambda$ should be put into the same branch if they have the same editing type. In Fig. 2, decreasing $\lambda$ to 0.8 groups "horse" and "field" into a single rigid edit branch, yielding slightly different results.
> > >
> > >
> > > ***Q3: How to control the editing strength for ParallelEdits?***
> > >
> > > ***A3:*** ParallelEdits can control the editing strength by adjusting the CFG (Classifier-Free Guidance). As Reviewer 8ae7 also noted, there's a trade-off between achieving satisfactory inversion and robust editing ability. A higher CFG tends to produce stronger editing effects but may lower inversion results and identity preservation. Currently, we have set a fixed CFG of 4.0 for our evaluations, as detailed in Section E, Implementation Details, in the Appendix.

---

> > > > ### Comment · Reviewer_6th7 · 2024-08-13
> > > >
> > > > Thanks for authors' reply. It is more clear for me.

---

> > > > > ### Author Response · Authors · 2024-08-13
> > > > > **Thanks for the reply!**
> > > > >
> > > > > We appreciate the reviewer’s constructive and positive feedback on our paper and responses and happy to see the reviewer find our responses clear. If satisfied, we respectfully ask if they would consider raising the final rating of our paper.

---

### Author Rebuttal · Authors · 2024-08-07

We thank all the reviewers for their time, insightful suggestions, and valuable comments. We are grateful for the positive recognition of the reviewers that our idea and task are interesting (Reviewers 8ae7 and J7sk), the method is efficient for application (Reviewers J7sk), and our editing results are impressive (Reviewers 6th7 and J7sk).

We have responded to each reviewer's comments in detail below. A PDF document has been uploaded to include additional figures and tables. We hope our response will address the reviewers' concerns.

---

### Decision · Program_Chairs · 2024-09-25

**Decision:**

Accept (poster)

**Comment:**

This submission received two weak accept and one borderline accept. This paper aims to make multiple objects or attributes editing, Reviewer 8ae7 and Reviewer J7sk agreed that simultaneously editing multiple objects or attributes is interesting and practical. Reviewer 6th7 recognized the novelty of the method. The editing results are impressive.
The reviewers raised some concerns like unclear method details and gave some useful suggestions. The authors well addressed reviewers' concerns. Given the strong recommendation from Reviewer 8ae7 & 6th7 and borderline accept recommendation from Reviewer J7sk, the AC recommends this paper for acceptance.